# Enzymes and cellular interplay required for flux of fixed nitrogen to ureides in bean nodules

Luisa Voß[1], Katharina J. Heinemann [1], Marco Herde[1], Nieves Medina-Escobar[1] & Claus-Peter Witte [1] ✉

Tropical legumes transport fixed nitrogen in form of ureides (allantoin and allantoate) over long distances from the nodules to the shoot. Ureides are formed in nodules from purine mononucleotides by a partially unknown reaction network that involves bacteroid-infected and uninfected cells. Here, we demonstrate by metabolic analysis of CRISPR mutant nodules of *Phaseolus vulgaris* defective in either xanthosine monophosphate phosphatase (XMPP), guanosine deaminase (GSDA), the nucleoside hydrolases 1 and 2 (NSH1, NSH2) or xanthine dehydrogenase (XDH) that nodule ureide biosynthesis involves these enzymes and requires xanthosine and guanosine but not inosine monophosphate catabolism. Interestingly, promoter reporter analyses revealed that *XMPP*, *GSDA* and *XDH* are expressed in infected cells, whereas *NSH1*, *NSH2* and the promoters of the downstream enzymes urate oxidase (UOX) and allantoinase (ALN) are active in uninfected cells. The data suggest a complex cellular organization of ureide biosynthesis with three transitions between infected and uninfected cells.

Unlike most other plants, legumes can access atmospheric $N_2$ through a symbiotic relationship with nitrogen-fixing bacteria of the Rhizobiaceae family which invade the plant root and induce the formation of root nodules. In infected nodule cells, the bacteria differentiate forming organelle-like structures called bacteroids which harbor the oxygen-sensitive nitrogenase complex for $N_2$ fixation to ammonium. The plant receives ammonium from the bacteroids in exchange for reduced carbon compounds and other nutrients[1]. Ammonium is assimilated by the plant and long-distance nitrogen transport metabolites are generated in the nodule for nitrogen export through the nodule vascular system to the shoot. Most legumes of temperate climates, such as *Medicago truncatula* or *Lotus japonicus*, use the amides glutamine and asparagine as transport compounds, whereas many legumes of (sub)tropical origin, which include important crops like *Glycine max* (soybean) or *Phaseolus vulgaris* (common bean), synthesize the ureides allantoin and allantoate[2,3]. The center of the determinate nodule of soybean and common bean contains larger infected cells, each making contact with interspersed smaller uninfected cells[4,5]. The uninfected cells form strands out of the central area into the surrounding inner cortex, which is composed of several layers of uninfected cells and contains the vascular system enclosed by an endodermis with a Casparian strip. The sclerenchyma, consisting of several layers of parenchyma cells, separates inner from outer cortex[6].

Ureides are made by de novo synthesis of purine nucleotides[2] followed by their partial degradation[7] (Supplementary Fig. 1). In plants, de novo purine nucleotide synthesis via inosine monophosphate (IMP) to adenosine monophosphate (AMP) is located in plastids[8] although in nodules of ureide-exporting *Vigna unguiculata* (cowpea) also a mitochondrial localization has been claimed[9]. Purine nucleotides are likely exported from the organelles to the cytosol because a cytosolic IMP dehydrogenase (IMPDH), that oxidizes IMP to xanthosine monophosphate (XMP, Supplementary Fig. 1), appears to be involved in ureide biosynthesis in cowpea nodules[10]. In cell-free experiments, IMP and XMP are usually both efficient precursors for ureides, whereas GMP

[1]Department of Molecular Nutrition and Biochemistry of Plants, Leibniz Universität Hannover, Herrenhäuser Straße 2, 30419 Hannover, Germany. ✉e-mail: cpwitte@pflern.uni-hannover.de

and especially AMP are not[10,11]. A clear intermediate of ureide bio-synthesis is the purine base xanthine, which accumulates in nodules when xanthine dehydrogenase (XDH) activity is compromised by allopurinol[12,13], an inhibitor of the enzyme, or when *XDH* is mutated[14]. In summary, these studies established that IMP and xanthine are inter-mediates of ureide biosynthesis and that in vivo IMP is likely oxidized to XMP, from which then the base is released by unknown enzymes. This model implies the existence of an XMP dephosphorylating enzyme generating xanthosine and of a xanthosine hydrolase or phosphorylase producing xanthine. That plants indeed possess an XMP phosphatase (XMPP) involved in purine catabolism in vivo has been recently demonstrated in *Arabidopsis thaliana*[15]. This cytosolic enzyme is conserved in plants. In Arabidopsis, xanthosine can also be derived from GMP dephosphorylation to guanosine and its deamina-tion by the cytosolic enzyme guanosine deaminase (GSDA)[16]. *GSDA* mutation strongly reduces the xanthosine pool[15,17] and leads to increased guanylate concentrations[18]. *P. vulgaris* has three very similar copies of *GSDA*, which have not yet been characterized. The degra-dation of xanthosine has also been investigated already in Arabidopsis. This plant possesses two cytosolic nucleoside hydrolase complexes, a nucleoside hydrolase 1 (NSH1) homomeric complex with prominent uridine hydrolase activity and a heteromeric complex formed by NSH1 and nucleoside hydrolase 2 (NSH2) which is highly active with xan-thosine and inosine. Because interaction of NSH1 with NSH2 is required for NSH2 activation, a mutation in *NSH1* abolishes the activities of both complexes[17]. These enzymes are conserved in plants[19,20] and orthologs from *P. vulgaris* have been characterized recently[21]. So far, a possible role of these different enzymes in generating xanthine from IMP for ureide biosynthesis in nitrogen-fixing nodules of tropical legumes has not been investigated.

Purine de novo synthesis probably occurs mainly in infected cells of cowpea nodules[22]. The cellular localization of the enzymes gen-erating xanthine from IMP have not yet been investigated in legumes, but they are thought to be located in infected cells[2,3]. Marked xanthine catabolism by XDH was found only in infected cells of several ureide-producing legumes by histochemical activity staining. Such staining was blocked by allopurinol and was not observed in nodules of amide-producing legumes[23]. By contrast, XDH was reported to reside mainly in uninfected cells of cowpea nodules using immunogold staining[24]. Thus, the cellular localization of XDH still requires further investiga-tion. However, it is clear that the product of XDH, urate, is oxidized to 5-hydroxyisourate exclusively in uninfected cells by urate oxidase (UOX)[25–27]. UOX resides in enlarged peroxisomes which mainly occur in uninfected cells in the infection zone and in the first three cortical cell layers surrounding the infection zone[28]. 5-hydroxyisourate is con-verted to *S*-allantoin by the peroxisomal allantoin synthase, an enzyme so far only investigated in Arabidopsis[29,30]. From the peroxisome, allantoin must be exported to the endoplasmic reticulum because further hydrolysis to allantoate by allantoinase (ALN) occurs in this organelle[31,32] in uninfected cells[33]. Mainly allantoate but also allantoin are used for long-distance nitrogen export from the nodules to the shoot[34–36]. Allantoin and allantoate migrate at least in part apoplasti-cally through the nodule cortex towards the vascular tissue because ureide permeases (UPS), which are ureide importers, are required for efficient ureide export from the nodules. UPS are located at the endodermis for upload into the symplast to overcome the Casparian strip and in the vascular tissue[37,38].

In this work, we established the CRISPR technology in hairy roots of *P. vulgaris* to create root and nodule mutants of *XMPP, GSDA, NSH1* and *NSH2* and *XDH* alone and in certain combinations. Targeted metabolic profiles focused on purine nucleotide catabolism were generated by liquid chromatography coupled to mass spectrometry (LC-MS) using nodules of these genetic variants to assess their possible roles in nodule ureide biosynthesis. Additionally, a comprehensive promoter-reporter study was conducted in transgenic nodules to investigate the cellular expression domains of the above-mentioned genes and of *UOX* and *ALN*. Using this data we created a revised model of ureide biosynthesis in nodules of *P. vulgaris*.

## Results

### Meta-analysis of transcription data from legumes

Our aim was to identify genes/proteins involved in ureide biosynthesis in nodules of tropical legumes using *P. vulgaris* as experimental sys-tem. It is known that the activities of enzymes involved in ureide bio-synthesis are correlated with nodule activity[35,39]. We therefore hypothesized that transcripts of genes participating in ureide pro-duction will be specifically upregulated in nodules compared to roots of tropical legumes, like common bean and soybean, but not in legumes from temperate climates, like Lotus and Medicago. From publically available transcriptome studies of the four mentioned legume species, we compiled expression data of known and candidate genes for ureide biosynthesis from roots and nodules. Ratios of nodule to root expression were calculated and listed if they were above 1.5 (Table 1). The data show that exclusively in tropical legumes all genes of purine nucleotide synthesis up to IMP are transcriptionally upre-gulated in nodules versus roots, whereas genes required for the synthesis of AMP from IMP (*ASS*) and the degradation of AMP to IMP (*AMPD*) are not upregulated. This strongly suggests that AMP is not an intermediate of ureide biosynthesis. All genes of IMP degradation to allantoate, mostly known from work in Arabidopsis, are upregulated almost exclusively in the nodules of tropical legumes—notably also the genes for the recently discovered XMP phosphatase (XMPP) and the nucleoside hydrolases NSH1 and NSH2 indicating that these are involved in ureide biosynthesis. Interestingly, the GMP synthetase gene (*GMPS*) is not upregulated and of the three *GSDA* genes only one is slightly induced. Thus, it seems that the branch via GMP and gua-nosine does not play a role for allantoin and allantoate synthesis in nodules although in Arabidopsis it is involved in the generation of ureides[15,17]. The ureide catabolic genes for allantoate amidohydrolase (AAH), ureidogylcine aminohydrolase (UGAH) and ureidoglycolate amidohydrolase (UAH), which together catalyze the full hydrolysis of allantoate to glyoxylate, carbon dioxide and ammonia[40–42], are also not induced. This is to be expected because allantoate would already be hydrolyzed in the nodule in the presence of these enzymes.

### Biochemical characterization of the XMP phosphatase from common bean

Tracer studies with radiolabeled nucleotides suggested that XMP dephosphorylation may play an important role for ureide production in nodules[10,11]. Very recently, it was reported that Arabidopsis pos-sesses an XMPP that catalyzes the entry reaction into purine nucleotide catabolism from XMP to xanthosine[15]. This enzyme is conserved in plants including *P. vulgaris* (Supplementary Fig. 2a) and the corre-sponding gene is transcriptionally induced in nodules (Table 1). For biochemical characterization, we transiently expressed a C-terminal Strep-tagged variant of the XMPP ortholog from *P. vulgaris* in *Nicoti-ana benthamiana* and affinity purified the enzyme (Supplementary Fig. 2b). The phosphatase has a high catalytic activity for XMP with a $K_M$ of $7.3 \pm 2.4\,\mu M$ and a turnover number of $18.4 \pm 1.7\,s^{-1}$ (Supple-mentary Fig. 2c), similar to the parameters obtained for the Arabi-dopsis ortholog ($K_M = 3.9 \pm 0.2\,\mu M$, $k_{cat} = 9.2 \pm 0.2\,s^{-1}$)[15].

### Generation of mutant nodules by CRISPR mutation in hairy roots

To investigate whether *XMPP* as well as *GSDA.1, GSDA.2, GSDA.3, NSH1* and *NSH2* are involved in ureide biosynthesis in *P. vulgaris* nodules, nodulated hairy roots expressing CRISPR transgenes targeting the respective genes were generated. *XDH* was also targeted as control because its importance for ureide biosynthesis is known[14]. Nodules expressing a construct not encoding sgRNAs were used as wild type

**Table 1 | Ratios of transcript abundances in nodules versus roots in ureide- and amide-exporting legumes for genes of purine nucleotide and ureide biosynthesis**

| Locus no.[a] | Annotation[b] | Pathway | Ratio of transcript abundances in nodules versus roots[c] | | | |
|---|---|---|---|---|---|---|
| | | | *P. vul.* | *G. max.* | *L. jap.* | *M. tru.* |
| (Phvul.nnn) | gene/protein | synthesis of | | | | |
| 003G228800 | PRS | IMP de novo | 52.5 | 16.9 | no | 4.6 |
| 009G002200 | Pur1, PRAT | IMP de novo | 19.3 | 509.0 | no | no |
| 007G182100 | Pur2, GARS | IMP de novo | 7.4 | 13.7 | no | no |
| 001G115300 | Pur3, GART | IMP de novo | 8.7 | 21.2 | no | no |
| 001G234500 | Pur4, FGARAT | IMP de novo | 11.6 | 15.5 | no | no |
| 002G273500 | Pur5, AIRS | IMP de novo | 10.9 | 5.0 | no | no |
| 007G188800 | Pur6, AIRC | IMP de novo | 6.9 | 9.9 | no | no |
| 008G178500 | Pur7, SAICARS | IMP de novo | 6.8 | 31.1 | no | no |
| 008G226900 | Pur8, ASL | IMP de novo | 10.7 | 15.7 | no | no |
| 005G018700 | Pur9/10, ATIC | IMP de novo | 8.2 | 10.0 | no | no |
| 001G136400 | ASS | AMP from IMP | no | no | no | no |
| 010G078200 | AMPD-1 | IMP from AMP | no | no | no | no |
| 003G070600 | AMPD-2 | IMP from AMP | no | no | no | no |
| 002G290100 | GMPS | GMP from IMP | no | no | no | no |
| 007G185600 | GSDA.1 | xanthosine | no | no | no | no |
| 009G220800 | GSDA.2 | xanthosine | 2.8 | 7 | no | no |
| 003G124100 | GSDA.3 | xanthosine | no | no | no | no |
| 007G056000 | XMPP | xanthosine | 16.5 | 60.7 | 1.8 | no |
| 005G048700 | IMPDH | XMP from IMP | 19.4 | 13.6 | no | no |
| 001G188700 | NSH1 | ureides | 6.8 | 12.7 | no | no |
| 003G000600 | NSH2 | ureides | 3.8 | 7.5 | no | no |
| 005G148000 | XDH | ureides | 6.7 | 4.4 | no | no |
| 007G234300 | UOX | ureides | 26.4 | 14.5 | 4.3 | no |
| 010G033700[d] | ALNS | ureides | 8.3 | 11.0 | no | no |
| 006G186700 | ALN | ureides | 40.9 | 18.3 | no | no |
| 009G242900 | AAH | NH$_3$ from ureides | no | no | no | no |
| 003G225600 | UGAH | NH$_3$ from ureides | no | no | no | no |
| 007G125700[d] | UAH | NH$_3$ from ureides | no | no | no | no |

[a]We used the induction of at least one paralog as an indicator of involvement in ureide biosynthesis. Therefore, in several cases not all paralogs are listed, but only those induced in nodules versus roots. Non-induced paralogs may also contribute to ureide biosynthesis.

[b]*PRS* phosphoribosylphosphate synthetase, *PRAT* phosphoribosyl pyrophosphate amidotransferase, *GARS* glycinamide ribonucleotide synthase, *GART* glycinamide ribonucleotide transformylase, *FGARAT* formylglycinamide ribonucleotide amidotransferase, *AIRS* aminoimidazole ribonucleotide synthetase, *AIRC* aminoimidazole ribonucleotide carboxylase, *SAICARS* succinoaminoimidazolecarboximide ribonucleotide synthetase, *ASL* adenylosuccinate lyase, *ATIC* AICAR transformylase and IMP cyclohydrolase, *ASS* adenylosuccinate synthetase, *AMPD* AMP deaminase, *GMPS* GMP synthetase, *GSDA* guanosine deaminase, *XMPP* XMP phosphatase, *IMPDH* IMP dehydrogenase, *NSH1* nucleoside hydrolase 1, *NSH2* nucleoside hydrolase 2, *XDH* xanthine dehydrogenase, *UOX* urate oxidase, *ALNS* allantoin synthase, *ALN* allantoinase, *AAH* allantoate amidohydrolase, *UGAH* ureidoglycine aminohydrolase, *UAH* ureidoglycolate amidohydrolase.

[c]Statistical analysis with the exact test for two-group comparison of Robinson and Smyth. Ratios below 1.5 are indicated with 'no' for not induced in nodules versus roots. *P. vul.* Phaseolus vulgaris, *G. max* Glycine max, *L. jap.* Lotus japonicus, *M. tru.* Medicago truncatula.

[d]gene codes are from ver1.0 of the of the P. vulgaris genome, all codes are identical to the current version (ver. 2.1) except for allantoin synthase and ureidoglycolate amidohydrolase, here the codes in ver2.1 are Phvul.010G033866 and Phvul.007G125750, respectively.

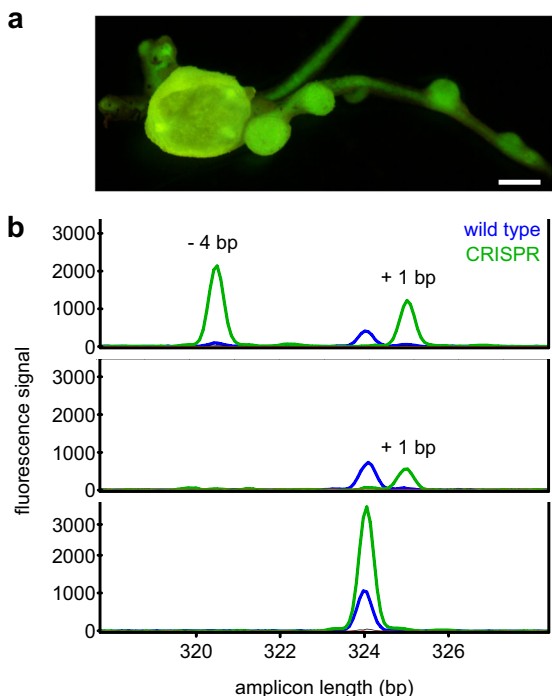

**Fig. 1 | Fluorescence of transformed roots and AFLP analysis of nodule pools. a** Transgenic hairy roots expressing GFP as selection marker. Images were taken with a binocular equipped with a GFP-long path filter. Scale bar, 500 µm. **b** AFLP analyses with capillary electrophoresis for mutant identification on three example chromatograms. Green peaks represent JOE-labelled amplicons from XMPP-CRISPR-transformed nodules. Blue peaks represent 6-FAM-labelled (wild type size) amplicons from control nodules transformed with an empty vector. From top to bottom, AFLP analysis of three independent nodule pools showing biallelic null-mutant, homozygous null-mutant and wild type patterns. Analysis was performed once per sample and the mutations of the samples identified as null-mutants were further confirmed by Sanger sequencing.

controls. The transformation vector contained a green fluorescent protein (GFP) expression cassette for selection of transformed nodules (Fig. 1a). Nodules from each transformed root were pooled, homogenized and freeze-dried. Each nodule pool represented one sample for mutation and metabolite analysis. Mutations due to insertions or deletions caused by CRISPR-mediated double strand breaks and error-prone repair by non-homologous end joining were detected by amplified fragment length polymorphism (AFLP) analysis (Fig. 1b). *P. vulgaris* is diploid, thus only samples with AFLP patterns consistent with frame-shift mutations in both alleles of each targeted gene were confirmed by Sanger sequencing and used for metabolite analysis. For all targeted genes except *GSDA.1*, several independent null-mutant nodule pools were obtained including the double and triple mutants *gsda.2 gsda.3* and *xmpp gsda.2 gsda.3* (Table 2). Note that editing of *GSDA.1* failed although transgenic nodules were easily obtained and the sgRNA encoded in the respective constructs was functional according to in vitro cleavage assays with Cas9 (Supplementary Fig. 3). The predictive power of in vitro sgRNA pretests for editing success in vivo is clearly not absolute. However, overall the chosen strategy for generating nodule mutants was efficient and usually resulted in several homozygous or biallelic null-mutants. Chimeric mutants were rare (Table 2) indicating that Cas9-induced double strand breaks and their repair occurred at the onset of hairy root development.

**Ureides are made via xanthosine generated from XMP or GMP**
Metabolites of purine catabolism were quantified in the different mutant and control nodule pools to clarify the pathway of ureide biosynthesis in nodules. Common bean nodules with a defect in *XDH*

**Table 2 | Overview of CRISPR-mediated mutations in target genes of *Phaseolus vulgaris***

| | C | XMPP | GSDA.1 | GSDA.2 | GSDA.3 | XMPP GSDA.2 GSDA.3 | GSDA.2 GSDA.3 | XDH | NSH1 | NSH2 |
|---|---|---|---|---|---|---|---|---|---|---|
| No. of pools | 8 | 15 | 15 | 15 | 15 | 41 | 15 | 23 | 14 | 24 |
| Wild type[a] | 8 | 4 | 15 | 6 | 3 | 7 | 4 | 10 | 11 | 21 |
| Chimeric mutant[a] | 0 | 0 | 0 | 0 | 1 | 2 | 1 | 1 | 0 | 0 |
| Incomplete mutant[a] | 0 | 6 | 0 | 6 | 8 | 28 | 7 | 9 | 0 | 0 |
| Null-mutant[a] | 0 | 5 | 0 | 3 | 3 | 4 | 3 | 3 | 3 | 3 |

[a]Samples were classified as null-mutant if both alleles of the gene were altered by an insertion or deletion that caused a frame-shift in the gene's open reading frame. If only one allele was changed in this way, the sample was classified as an incomplete mutant (including cases of biallelic mutations with at least one allele maintaining an intact open reading frame). Only few samples showed a chimeric mutation pattern (more than two peaks in AFLP different from the wild type amplicon). These were omitted from further analysis. C, control (nodules transformed with construct not encoding sgRNAs).

accumulated high amounts of xanthine and did not produce ureides since allantoin and allantoate were both undetectable[14] (Fig. 2a). In leaves of Arabidopsis *xdh* plants, xanthine aggregates to fluorescent particles in vivo[43]. In cross sections of *xdh* nodules, black dots were observed in uninfected cells using bright field microscopy. These were fluorescent and are probably such xanthine-containing crystals (Fig. 2b). We also observed similar precipitates in uninfected cells of *xdh* nodules from soybean (Supplementary Fig. 4). The hypoxanthine concentration in common bean nodules was generally very low and did not increase in *xdh* background compared to wild type, although hypoxanthine is also an XDH substrate (Supplementary Fig. 1). It has been noted before that inhibition of XDH by allopurinol does not lead to an accumulation of hypoxanthine in nodules[12,13]. Also in Arabidopsis *xdh* lines, hypoxanthine concentrations remained similar to wild type[17]. The data suggest that xanthine is not primarily produced by IMP degradation via inosine and hypoxanthine in nodules and perhaps in plants in general.

XMP accumulates in *xmpp* and *xmpp gsda.2 gsda.3* nodules and is not detectable in nodules of other genotypes demonstrating that XMPP dephosphorylates XMP in vivo (Fig. 2a). In comparison to wild type nodules, the average allantoin and allantoate concentrations of *xmpp* nodules were reduced by about 50%—although this was statistically significant only for allantoate. However, the concentration of allantoate was by far higher than that of allantoin, thus allantoate is the main product of nodule ureide production. The data demonstrate that XMPP is involved but not indispensable for ureide biosynthesis. An alternative pathway runs from XMP to GMP via guanosine to xanthosine (Supplementary Fig. 1). The gene for GMP phosphatase is unknown and therefore cannot be genetically manipulated, but *GSDA* can. Unfortunately, we were unable to mutate all three *GSDA* isogenes but the available *GSDA* mutants had less allantoate than the control showing that this route contributes to ureide biosynthesis (Fig. 2a). There is also evidence that the flux through this route is increased in *xmpp* background, because the guanosine content is elevated in *xmpp* nodules compared to wild type. This effect is even more evident when comparing *gsda.2 gsda.3* nodules with *xmpp gsda.2 gsda.3* nodules, especially when additionally guanine concentrations are considered. Guanosine and guanine concentrations are correlated and, as in Arabidopsis[15], the connection is probably made by the NSH1/NSH2 complex (see below). It appears that blockage of the XMPP route leads to increased conversion of XMP to GMP which can be degraded via guanosine to xanthosine partially compensating for the loss of XMPP. The route to xanthosine via GMP is energetically more costly[15] but it may be unavoidable because GMPS, the enzyme catalyzing XMP to GMP amination, must be active in all cells to ensure guanylate supply. It is possible that GMPS generates too much GMP when the flux through XMP is high for ureide production. In agreement with this model, the mRNA of *GMPS* is not induced in nodules and from the *GSDAs* only *GSDA.2* is slightly induced (Table 1) indicating that there is no particular upregulation of the GMP-guanosine branch for the purpose of

ureide production. The slight *GSDA.2* induction may just provide enough GSDA capacity to remove the products of unavoidable GMP overproduction.

Interestingly, xanthosine also accumulates in *xmpp* nodules, as does inosine, although only slightly. We speculate that this is due to an inhibition of xanthosine and inosine hydrolase activity of the NSH1/NSH2 complex by the strongly increased XMP or guanosine concentrations in *xmpp* nodules.

In summary, the data show that ureides in nodules are made by the XMP-xanthosine pathway and also by the GMP-guanosine route but not via inosine and hypoxanthine.

### A NSH1/NSH2 complex has enhanced xanthosine hydrolase activity and is inhibited by guanosine

The nucleoside hydrolases NSH1 and NSH2 might also be involved in ureide biosynthesis. It has been shown in Arabidopsis that *At*NSH1 is mainly a uridine hydrolase with some xanthosine/inosine hydrolase activity. However, the catalytic efficiency for xanthosine of a complex of *At*NSH1 with *At*NSH2 is over 70-fold higher than for *At*NSH1 alone which was mainly attributed to *At*NSH2 activity[17]. As both enzymes are conserved at the C-terminus (Supplementary Fig. 5)[19] they were tagged at the less conserved N-terminus in the Arabidopsis study. Interestingly, *At*NSH2 was activated only in complex with *At*NSH1 both in vitro and in vivo and irrespective of whether it was tagged or not[17]. The orthologous nucleosidases from *P. vulgaris* were also recently characterized after expression of C-terminal tagged variants in *E. coli* and *N. benthamiana*. It was found that, in contrast to the Arabidopsis enzymes, NSH1 and NSH2 do not interact with each other and that NSH2 is active on its own with xanthosine[21].

We suspected that a C-terminal tag might interfere with the interaction and activation properties of the nucleoside hydrolases. Therefore, we reexamined the potential interaction of the NSHs from Phaseolus with N-terminal Strep- and myc-tagged variants. These were transiently co-expressed in *N. benthamiana* leaves and the proteins purified via the Strep tag. Because myc-tagged NSH1 could be co-purified with Strep-tagged NSH2 and vice versa (Fig. 3), we conclude that NSH1 and NSH2 from Phaseolus can interact. Additionally, NSH1 can interact with itself whereas NSH2 cannot. The specific activities of affinity purified NSH1, NSH2 and the NSH1/NSH2 complex were measured using 0.125 mM xanthosine as substrate. NSH1 but not NSH2 showed activity with xanthosine and the activity of the NSH1/NSH2 complex was about 12-fold higher than that of NSH1 (Table 3). These results are similar to those obtained in the detailed analysis of the Arabidopsis enzymes[17].

Nodules with mutated *XMPP* accumulate xanthosine (Fig. 2a) possibly caused by elevated concentrations of XMP or guanosine in this genetic background inhibiting xanthosine nucleoside hydrolase activity. We therefore examined the enzymatic activities of the purified nucleoside hydrolases with xanthosine in the presence of the potential inhibitors. The activities were not changed by XMP up to 0.125 mM, but

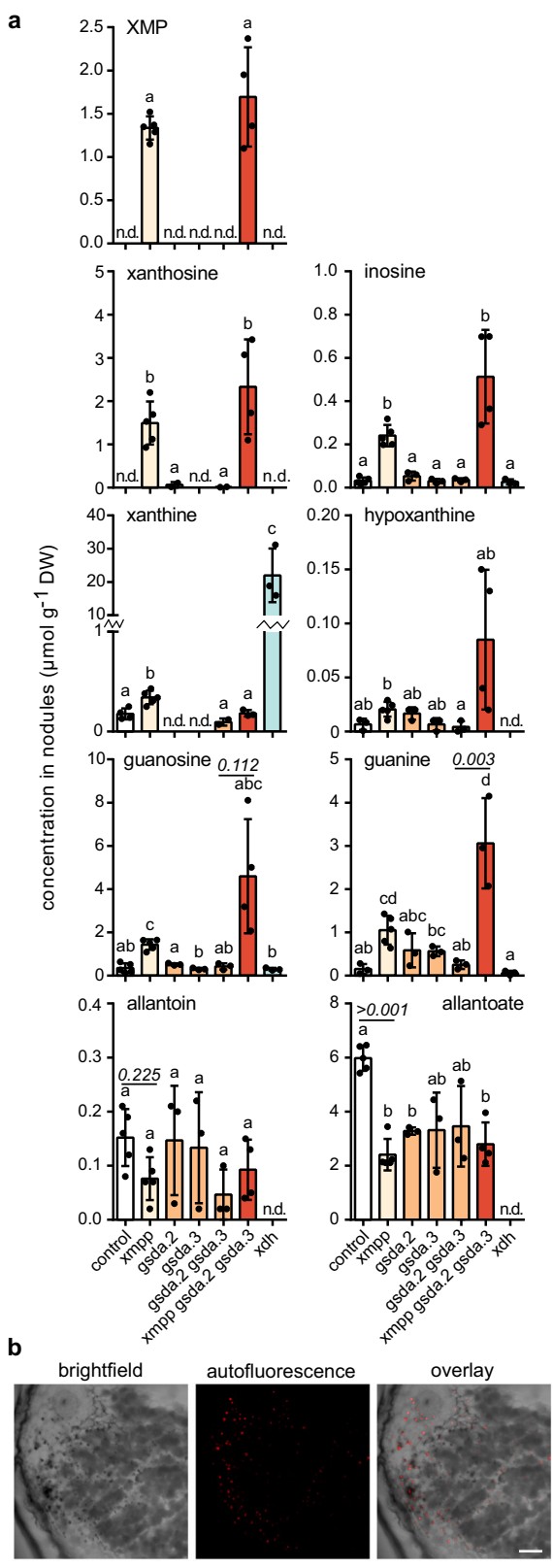

**Fig. 2 | Metabolic alterations in nodules of *XMPP*, *GSDA* and *XDH* genetic variants and fluorescent bodies in *xdh* nodules. a** Quantification of metabolites of purine catabolism and ureide synthesis in nodule extracts. Error bars are SD, measure of the center is the mean value, $n = 5$ for control and *xmpp*, $n = 4$ for *xmpp gsda.2 gsda.3*, $n = 3$ for *gsda.2*, *gsda.3*, *gsda.2 gsda.3* and *xdh*. A repeat ($n$) is a pool of nodules from a single transgenic root. Although different mutant roots / nodules can come from the same plant, in this experiment all null-mutant nodules came from roots of independent plants grown together. Statistical analysis with two-sided Tukey's pairwise comparison using the sandwich variance estimator. Different letters indicate $p$ values < 0.05. Selected $p$ values are indicated, all $p$ values can be found in the Source Data file. DW dry weight. **b** From left to right, brightfield image of an xdh nodule cross section with bacteroid infected (dark grey) and uninfected (light grey) cells and precipitates (black dots); confocal fluorescence image of the same cross section with autofluorescent objects (excitation 552 nm, emission 595–622 nm); overlay of both images. Scale bar, 100 μm. Image is representative of at least three nodules from independent transgenic roots. The experiment was performed once in Phaseolus and once in Soybean (Supplementary Fig. 4b, d, e) with similar results.

(Table 2) and metabolite contents analyzed. Since NSH2 requires interaction with NSH1 for activation, a mutant in *NSH1* can be considered a functional knockout of NSH1 and NSH2. Nodules lacking *NSH1* accumulated xanthosine and had a strongly reduced allantoate content (Fig. 4) demonstrating that NSH1 is a critical enzyme for ureide biosynthesis. Such nodules also contained more guanosine (i) because xanthosine is known to partially inhibit XMPP leading to an efflux of XMP into the guanylate pool[15] (see also guanosine in *xmpp* in Fig. 2) (ii) because xanthosine might also partially block GSDA by product inhibition (iii) because NSH activity for guanosine hydrolysis is absent—although this activity is weak[21], its absence causes an increased guanosine/guanine ratio of about 5 in *nsh1* (Fig. 4) versus <2 in wild type or *xmpp* (Fig. 2a). NSH1 and especially the NSH1/NSH2 complex also have inosine hydrolase activity[17,21] explaining the slightly elevated inosine content in *nsh1*. There is about 40-fold less inosine than xanthosine + guanosine in *nsh1*. This suggests that ureides are not formed in vivo via the IMP-inosine-hypoxanthine pathway, which was also indicated by the results with *xdh* nodules (Fig. 2b).

Interestingly, metabolite contents in *nsh2* nodules did not differ from wild type. Although this suggests that NSH2 is not required for ureide biosynthesis in nodules, our biochemical data show that NSH1 and NSH2 interact and that the complex has significantly stronger xanthosine hydrolase activity than NSH1 alone (Table 3). *NSH2* is also induced in nodules of tropical legumes (Table 1). Nonetheless, the xanthosine hydrolase activity of NSH1 appears to be sufficient to prevent xanthosine accumulation which resembles the situation in Arabidopsis. However, it was shown in Arabidopsis that plants expressing inactive NSH1 also did not accumulate xanthosine because NSH2 could still be activated by the defective interaction partner[17] suggesting that both activities need to be removed to observe an effect on metabolite level.

Despite the central role of NSH1, ureide biosynthesis is not completely inhibited in *nsh1* nodules, in contrast to *xdh* nodules. In Arabidopsis, this 'leakiness' of *NSH1* mutants as well as of *NSH1 NSH2* double mutants was also observed and was explained with a possible non-enzymatic hydrolysis of xanthosine or an enzymatic side activity which may in part be fostered by the high accumulation of the metabolite in these genetic backgrounds[17].

In summary, our results demonstrate that NSH1 is a central player for ureide biosynthesis. Taking into account all available data, it seems likely that NSH1 does not act alone but in an NSH1/NSH2 complex.

the addition of 0.25 mM guanosine reduced the activity of the NSH1/NSH2 complex about twofold and the activity of NSH1 could no longer be detected (Table 3).

## NSH1 is involved in ureide biosynthesis

To assess the involvement of NSH1 and the NSH1/NSH2 complex in nodule ureide biosynthesis, *nsh1* and *nsh2* nodules were generated

## Ureide biosynthesis requires the interplay of different nodule cell types

Ureide biosynthesis is initiated in infected cells and eventually shifts to uninfected cells, but the point of transition is still unclear. We used a

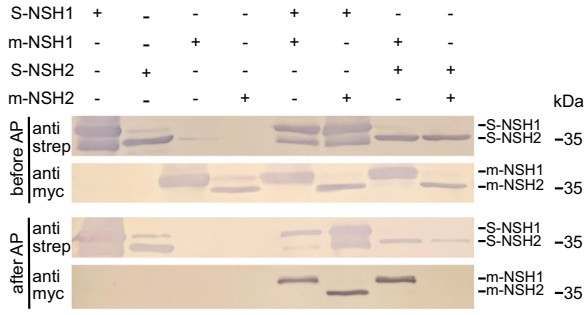

**Fig. 3 | Interaction of NSH1 and NSH2 from Phaseolus vulgaris.** NSH1 and NSH2 were transiently (co)-expressed as myc- or Strep-tagged variants in *N. benthamiana*. N-terminal tagged proteins were affinity purified using the Strep tag. Protein expression (top two panels) and protein (co)-purification by affinity chromatography (bottom two panels) were assessed by immunodetection with either Strep-Tactin alkaline phosphatase conjugate (anti strep) or anti c-myc (anti myc) antibodies. 10 µl of clarified leaf extracts or affinity-purified proteins were loaded. AP affinity purification, prefix m- c-myc-tagged, prefix S- Strep-tagged. The experiment was performed three times with similar results.

**Table 3 | Specific activities of common bean NSH enzymes alone and in complex**

| Specific activity[a, b] (µmol min$^{-1}$ mg$^{-1}$) | | | |
| --- | --- | --- | --- |
| Xanthosine (mM) | 0.125 | 0.125 | 0.125 |
| Guanosine (mM) | – | 0.125 | 0.250 |
| Only NSH1 | 1.82 ± 0.06[c] | 0.84 ± 0.11 | n.d. |
| Only NSH2 | n.d. | n.d. | n.d. |
| NSH1/NSH2 complex[c] | 21.77 ± 0.50 | 17.19 ± 0.55 | 11.00 ± 2.03 |

[a]The enzymes were Strep- or myc-tagged at the N-terminus.
[b]Errors are SD (*n* = 3 technical replicates). All activity values were statistically different with *p* < 0.05 (see Source Data file).
[c]The NSH1/NSH2 complex was purified via Strep-tagged NSH2.

promoter-reporter system to investigate the promoter activity of several known and here newly described ureide biosynthetic genes in nodule cells as a proxy for the likely cellular location of the corresponding enzymes.

*P. vulgaris* seedlings were transformed with *A. rhizogenes* harboring constructs with promoters for the gene of interest driving the expression of a peroxisome-targeted mNeonGreen reporter. The peroxisomal targeting was chosen to increase the sensitivity by concentrating the reporter in a small organelle. Cross sections of nodules expressing the reporter gene under control of the promoters p*XMPP*, p*GSDA.1*, p*GSDA.2*, p*GSDA.3*, p*NSH1*, p*NSH2*, p*XDH*, p*UOX* or p*ALN* were analyzed using confocal fluorescence microscopy. Control nodules with a p35S-mNeonGreen transgene displayed fluorescence in all cell types except the bacteroid infected cells (Supplementary Fig. 6) indicating that p35S is not active there.

With p*XMPP* and all three *GSDA* promoters, mNeonGreen fluorescence was observed in the central region of the nodule, exclusively in infected cells (Fig. 5, Supplementary Fig. 6 and 7). For p*GSDA.3*, the signal was faint (Supplementary Fig. 6) which is in agreement with the relatively low abundance of mRNA from this gene in public RNAseq datasets[44]. The data indicate that XMP dephosphorylation and guanosine deamination occur in infected nodule cells. Curiously, mNeonGreen was not observed in punctate structures as would be expected for peroxisomes, although the same constructs, when transiently expressed in *N. benthamiana*, resulted in punctuate, likely peroxisomal localization of the marker (Supplementary Fig. 8). The promoters of *NSH1* and *NSH2* were both active exclusively in uninfected cells closely associated to infected cells, as mNeonGreen was only detected there (Fig. 5, Supplementary Fig. 7). The data suggest that ureide biosynthesis takes place in infected cells until the synthesis of xanthosine, which is then hydrolyzed to ribose and xanthine in neighboring uninfected cells by the NSH1/NSH2 complex. This is in agreement with the xanthine precipitates observed in these cells in *xdh* nodules (Fig. 2b, Supplementary Fig. 4). Surprisingly, promoter activity of *XDH*, required for the next step in ureide biosynthesis, was observed only in infected cells again and in this case the reporter was located in punctuate structures (Fig. 5, Supplementary Fig. 7 and 8). In this experiment, a 735 bp region between the stop codon of the upstream gene and the start codon of the *XDH* gene had been chosen as *XDH* promoter. Because the localization of the reporter in infected cells was unexpected, we additionally cloned a 3 kb fragment upstream of the XDH start codon into our reporter construct and assessed the transcriptional activity of this longer promoter fragment, which also

included part of the upstream gene (Fig. 6a). Both, the 735 bp and the 3 kb promoter fragments of *XDH* drove expression of the reporter only in infected cells (Fig. 6b, c). Rarely, fluorescence signals were apparently observed in uninfected cells, but one must take into account that the images were taken from unfixed 60-µm nodule slices, so some fluorescence may have originated from deeper cell layers or been displaced by the cutting process. In agreement with previous reports that located the next enzyme, UOX, in uninfected cells[25–27], we observed *UOX* promoter activity in uninfected cells closely associated to infected cells. According to these findings, urate produced by XDH in infected cells is transferred for its further oxidation by UOX to the uninfected cells in the infection zone. When expressing mNeonGreen under the control of p*ALN*, the fluorescent protein was located in cells surrounding the vascular bundles in the inner cortex of the nodule (Fig. 5, Supplementary Fig. 7). That ALN activity is probably mainly associated with uninfected cells has been reported before[33], but where exactly ALN is expressed has not yet been shown.

## Discussion

We have integrated our data into an updated model of ureide biosynthesis in the nodules of *P. vulgaris* (Fig. 7). Fixed nitrogen is released as ammonia by the bacteroids and is incorporated by purine biosynthesis into IMP in the infected cells. AMP is probably not an intermediate of ureide biosynthesis (Table 1). The main biosynthetic route for the ureides begins with the oxidation of IMP to XMP by IMPDH[10] followed by the hydrolysis of XMP to phosphate and xanthosine by XMPP in the infected cells. Because every cell needs GMP, some XMP is always aminated to GMP by GMPS. We hypothesize that in infected cells flux through GMPS is higher than required for GMP homeostasis because the XMP level is elevated there. Surplus GMP is hydrolyzed by an unknown phosphatase to phosphate and guanosine, which is deaminated to xanthosine by GSDA in the infected cells (Fig. 7). This XMPP bypass is costly, resulting in the hydrolysis of three phosphoanhydride bonds driving the GMPS reaction and the re-assimilation of ammonia released by GSDA[15]. The genes of the GMP-guanosine branch of xanthosine production are not (strongly) induced in nodules compared to roots (Table 1), indicating that although this route contributes to ureide biosynthesis (Fig. 2), it is not the main pathway. Xanthosine is exported from the infected cells to neighboring uninfected cells for hydrolysis to ribose and xanthine by the NSH1/NSH2 complex. Xanthine re-enters the infected cells for oxidation by XDH to urate, which again leaves the infected cells to neighboring uninfected cells for further oxidation by UOX in the peroxisomes. This seemingly complicated setup may be advantageous for two reasons: (i) If the NSH1/NSH2 complex were located in the infected cells, it would likely be exposed to higher guanosine concentrations, because the GMP-guanosine bypass is operative in these cells. Guanosine inhibits the NSH1/NSH2 complex[15] (Table 3) but is also slowly hydrolyzed by these enzymes to ribose and guanine[21]. However, guanine is not an intermediate of purine catabolism[17] and must be salvaged to GMP to re-

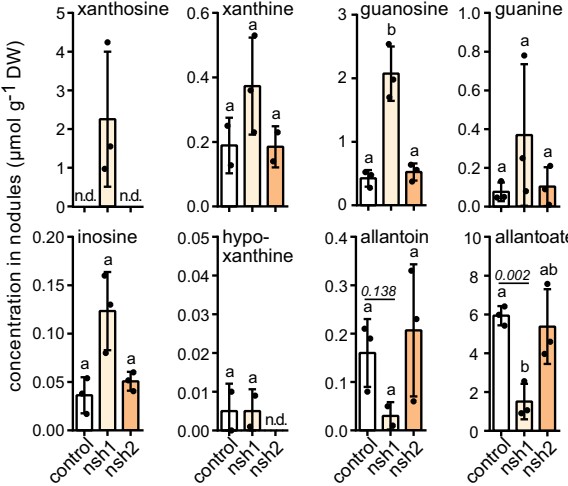

**Fig. 4 | Metabolic alterations in nodule mutants of *NSH1* and *NSH2*.** Quantification of metabolites of purine catabolism and ureide biosynthesis in *nsh1*, *nsh2* and control nodule extracts. Error bars are SD, measure of the center is the mean value, *n* = 3 for control, *nsh1* and *nsh2*. A repeat (*n*) is a pool of nodules from a single transgenic root. Although different mutant roots / nodules can come from the same plant, in this experiment all null-mutant nodules came from roots of independent plants grown together. Statistical analysis with two-sided Tukey's pairwise comparison using the sandwich variance estimator. Different letters indicate *p* values < 0.05. Selected *p* values are indicated, all *p* values can be found in the Source Data file. DW dry weight.

enter metabolism, which requires metabolic energy[8]. (ii) Xanthine made in the cytosol of uninfected cells enters the infected cells possibly through the symplast via plasmodesmata[5,45] encountering the cytosolic XDH[46] directly at the cell periphery. There xanthine is oxidized to uric acid, which is a potent scavenger for ROS[47,48] and reactive nitrogen species (RNS)[49]. Bean XDH activity itself is inhibited by RNS and is protected by urate[50]. A source of ROS are mitochondria that are particularly large and abundant at the cell periphery of infected cells[51] presumably to support energy-intensive nitrogen fixation and assimilation. Thus, urate production by XDH at the interface between infected and uninfected cells may create a sink for ROS/RNS in this zone of high ROS production and low ROS tolerance[52,53]. By contrast, the peroxisomal UOX reaction requires oxygen, providing a rationale for the presence of UOX in uninfected cells where oxygen is better available. Oxygen consumption by UOX may serve to further lower the oxygen exposure of the infected cells. UOX initiates a series of three peroxisomal reactions with unstable intermediates yielding *S*-allantoin[29,54], thus, allantoin is probably made in the cells where UOX is located. Interestingly, the promoter of the allantoin-degrading enzyme, ALN, is highly active in cells of the vascular tissue and not in the infection zone (Fig. 5) suggesting that allantoin hydrolysis to allantoate occurs preferentially there. This is supported by a high and locally focused concentration of allantoate in the inner cortex[55]. The *ALN* expression pattern probably creates an allantoin gradient from the uninfected cells of the infection zone to the vascular tissue enhancing directed allantoin diffusion towards the vasculature. Ample symplastic connections between uninfected cells in the infection zone through the inner cortex into the cells of the vascular tissue[5,45] allow for symplastic transport of allantoin. However, there is also evidence that apoplastic diffusion of allantoin plays a significant role. The allantoin importer UPS1 is strongly expressed at the vascular endodermis where a Casparian strip blocks the apoplastic diffusion pathway[37,38] and downregulation of UPS1 leads to ureide accumulation and defects in N partitioning from nodules to the shoot[38], whereas overexpression enhances these processes[56]. Allantoinase resides in the ER[31,32], thus allantoate will be produced there. Currently it is unclear how allantoate

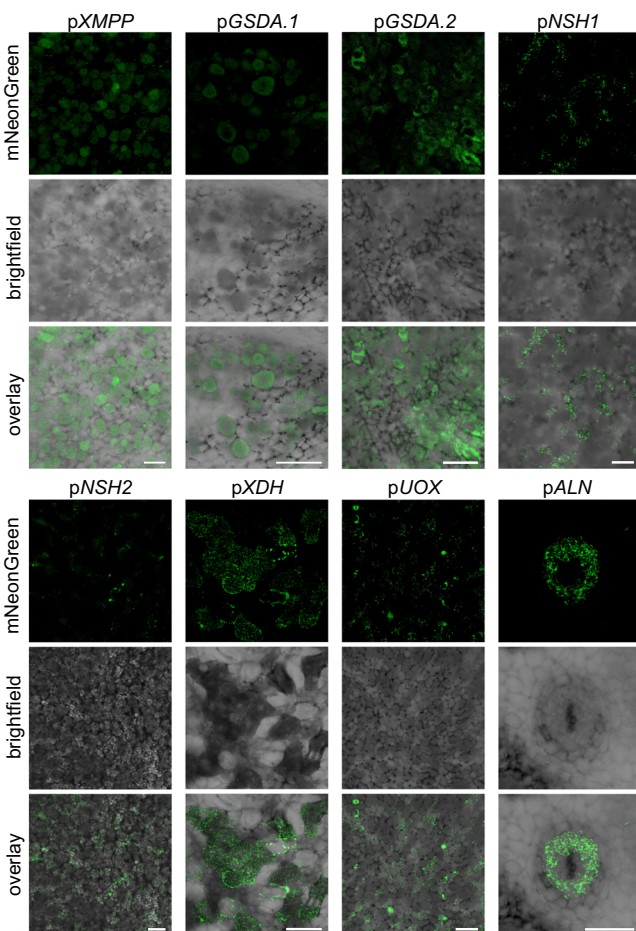

**Fig. 5 | Cell-type specific activity of ureide biosynthesis gene promoters in the common bean nodule.** Confocal fluorescence microscopy images of cross sections from common bean nodules expressing the coding sequence of peroxisome-targeted mNeonGreen under the control of native promoters of genes involved in ureide biosynthesis. The central infected region, consisting of infected (dark grey) and uninfected (light grey) cells, or the inner cortex with vascular bundle cells (for p*ALN*) are shown. From top to bottom, mNeonGreen channel, brightfield channel and overlay of both channels. Scale bar, 100 μm. Images are representative of at least three nodules from independent transgenic roots. The experiment was performed twice with similar results.

is exported from the ER to the xylem vessels. In our model, we have added a purely speculative element suggesting that this transport may be mediated by vesicular traffic from the ER to the plasma membrane.

## Methods
### Cloning
For each sgRNA tested in the Cas9 in vitro cleavage assay, a forward and a reverse primer were designed containing the DNA coding for the guide (20 bp) and a specific overhang of 4 bp (GAGG forward; AAAC reverse). For sgRNA 1, 2, 3 and 4, the specific primer pairs were P2193 and P2194, P2195 and P2196, P2197 and P2198, and P2199 and P2200, respectively. The complimentary primers (200 μM each) were incubated in a thermoblock at 95 °C in 10 mM Tris (pH 7.5), 50 mM NaCl and 1 mM EDTA. After 5 min, the thermoblock was turned off and the samples were incubated for an additional hour. The annealed primers were used for *Bbs*I cut-ligation into a modified version of pEn-Chimera. pEn-Chimera was a gift from Holger Puchta (Addgene plasmid #61432). Guide arrays were amplified from the assembled vectors using the primer pair P1997 and P1998. The amplicons were used as DNA template for in vitro transcription of the guide array with T7-RNA polymerase (ThermoFischer) according to the manufacturer's

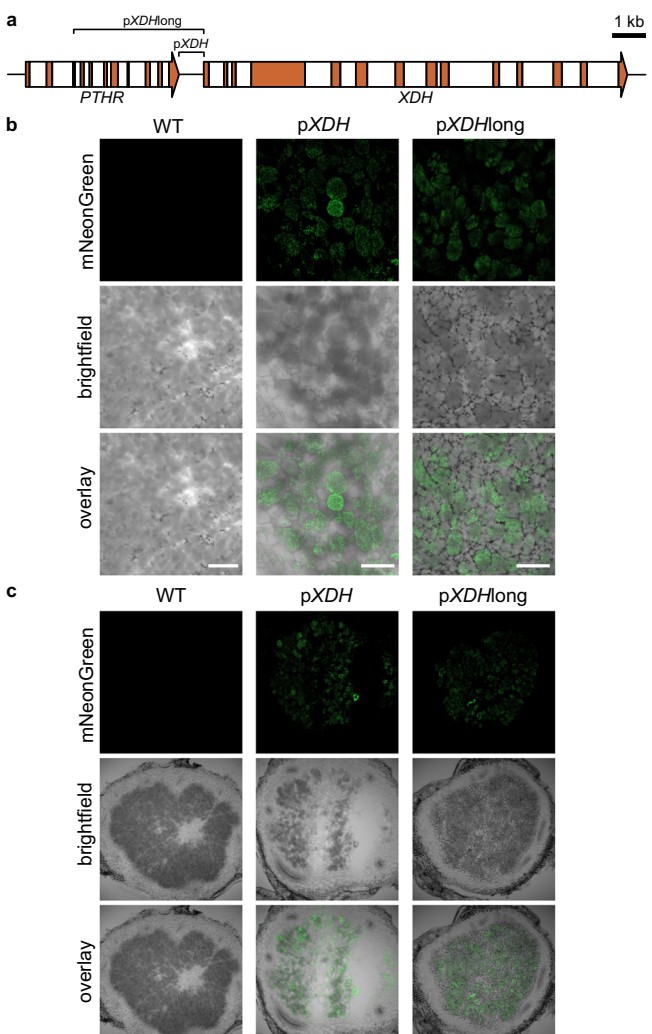

**Fig. 6 | Cell-type specific activity of the *XDH* promoter in the common bean nodule.** Confocal fluorescence microscopy images of cross sections from wild type (WT) nodules or nodules expressing the coding sequence of peroxisome targeted mNeonGreen under the control of the 735 bp *XDH* promoter (p*XDH*) or a 3 kb version of the promoter (p*XDH*long), that includes the 3'-part of a gene encoding a phosphate acyltransferase (PTHR; locus Phvul.L009443). **a** Schematic overview of gene structures and promoter region. Exon and intron structures within the coding region are displayed in orange and white, respectively. **b** Focus on the central infected region, consisting of infected (dark grey) and uninfected (light grey) cells. Scale bar, 100 μm. **c** Whole nodules. Scale bar, 225 μm. From top to bottom, mNeonGreen channel, brightfield channel and overlay of both channels. Images are representative of at least three nodules from independent transgenic roots. The experiment was performed once with the 3 kb promoter and three times with the 735 bp promoter, with similar results (Fig. 5 and Supplementary Fig. 7).

recommendations. The genomic sequence of *GSDA.1* was amplified using the primer pair P1485 and P1286. The amplicon was ligated into pJET1.2 (ThermoFischer) to generate the plasmid H1493, which was used as DNA template in the assay.

Vectors for induction of CRISPR/Cas9-mediated mutations in *Phaseolus vulgaris* were constructed by GoldenGate-cloning using the MoClo system[57,58]. The MoClo Toolkit and the Plant Parts Kit were donated from Nicola Patron and Sylvestre Marillonnet (Addgene kit #1000000044 and Addgene kit #1000000047). First, a variety of intermediate vectors were created. Turbo-GFP was amplified using the primer pair P626 and P627 (Supplementary Table 1) from pICSL50016 and cloned into pICH41308 after *Bbs*I cut-ligation, resulting in the generation of pICH41308_tGFP (V134). Combination of pICH47732,

pICH51266, V134 and pICH41421 in a *Bsa*I cut-ligation created pICH47732_35 S::tGFP::nos-t (V135). The primer pair P2250 and P2251 was used to amplify the ubiquitin promoter of parsley and the 5'-Cow Pea Mosaic Virus (CPMV) enhancer from V112[59]. The amplicon was cut with *Bbs*I and ligated into pICH41295 to create pICH41295_UBQ:5'CPMVenhancer (H987). The primer pair P824 and P825 was used to amplify the 3'-CPMV enhancer together with the 35 S terminator sequence from pXCScpmv-HAStrep (V69)[60]. The amplicon was digested with *Bbs*I and ligated into pICH41276 to create pICH41276_3'CPMVenhancer:35S-t (V157). Combination of pICH47742, H987, V150[61] and V157 in a *Bsa*I cut-ligation lead to the generation of pICH47742_UBQ:5'CPMVenhancer::Cas9::3'CPMVenhancer:35S-t (H1031). The CRISPOR web tool[62] was used for the sgRNA design and evaluation of potential off-target effects. The sgRNAs were chosen to bind in an exon and preferably within the first half of a corresponding transcript (Supplementary Fig. 9). Separate parts of the guide arrays were amplified from pGTR[63] using the long forward P293 or reverse primer P272, in combination with a guide specific forward or reverse primer as described by Xie[63]. The guide specific forward and reverse primer for *XMPP, GSDA.1, GSDA.2, GSDA.3, NSH1, NSH2* and *XDH* were P2325 and P2331, P2310 and P2311, P2312 and P2313, P2314 and P2315, P2777 and P2778, P2779 and P2780 and P2329 and P2332, respectively. To create the array, the separate parts were fused in a *Bsa*I cut-ligation. The arrays were re-amplified using the short forward and reverse primers P294 and P274 and ligated into a MoClo compatible shuttle vector[61] after *Bbs*I restriction digest. A first cut-ligation with *Bsa*I into pICH47751, resulting in a vector containing the specific guide arrays. These were further used in *Bbs*I cut-ligations with pAGM4723, V135, H1031 and pICH41766, leading to the following vectors for inducing CRISPR mutations: H1040 (directed against *XMPP*), H1041 (*XDH*), H1043 (*GSDA.1 GSDA.2 GSDA.3*), H1045 *(XMPP GSDA.1 GSDA.2 GSDA.3)*, H1351 (*NSH1*) and H1352 (*NSH2*). To generate the vector lacking a guide array, the final cut-ligation was performed combining pAGM4723, V135, H1031 and pICH41744, resulting in H1078.

For construction of vectors for promoter studies in bean mediated by *Agrobacterium rhizogenes* (*Rhizobium rhizogenes*) transformation and transient or stable transformation of plants mediated by *Agrobacterium tumefaciens* (*Rhizobium radiobacter*), the binary vector pAGM4723 conferring kanamycin resistance to bacteria was used as backbone. pAGM4723 was digested with *Bbs*I and *Pme*I to remove the multiple cloning site and the annealed primers P1481 and P1482 were inserted. The resulting vector (V193) was cut with *Pme*I and *Nde*I replacing a 652 bp insert with a PCR amplicon from pAGM4723 flanked by the same restriction sites generated with primers P1541 and P1542. Into the *Pme*I site of the resulting plasmid (V195) the annealed primers P1543 and P1544 encoding a right border, five repeats of a transfer stimulating sequence and an overdrive sequence were ligated. This generated pY2empty (V196). A synthetic mNeonGreen gene with an intron and encoding a C-terminal peroxisomal targeting sequence 1 (SKL) flanked by *Eco*RI and *Xba*I was cloned into pXCS-YFP[16] (V36) replacing the YFP coding sequence with the mNeonGreenSKL sequence. This generated pXCS-mNeonGreenSKL (V120). The expression cassette including 35 S promoter, mNeonGreenSLK coding sequence and 35 S terminator was excised from V120 by *Asc*I and *Not*I and ligated into pY2empty (V196) cut with *Asc*I and partially with *Not*I at the *Not*I site right next to the *Asc*I site. This produced pY2CS-mNeonGreenSKL (V197, documented in the Source Data file).

Putative promoter and 5'-UTR regions upstream of the AUG start codon of *XMPP, GSDA.1, GSDA.2, GSDA.3, NSH1, NSH2, XDH, UOX* and *ALN* were amplified with the primer pairs P1491 and P1492, P1497 and P1498, P1501 and P1502, P1505 and P1506, P1509 and P1510, P1513 and P1514, P1515 and P1516, P1517 and P1518, P1521 and P1522, respectively, introducing flanking *Asc*I and *Xho*I restriction sites. Usually the promoter sequences had a length of 3 kb or extended from the start codon

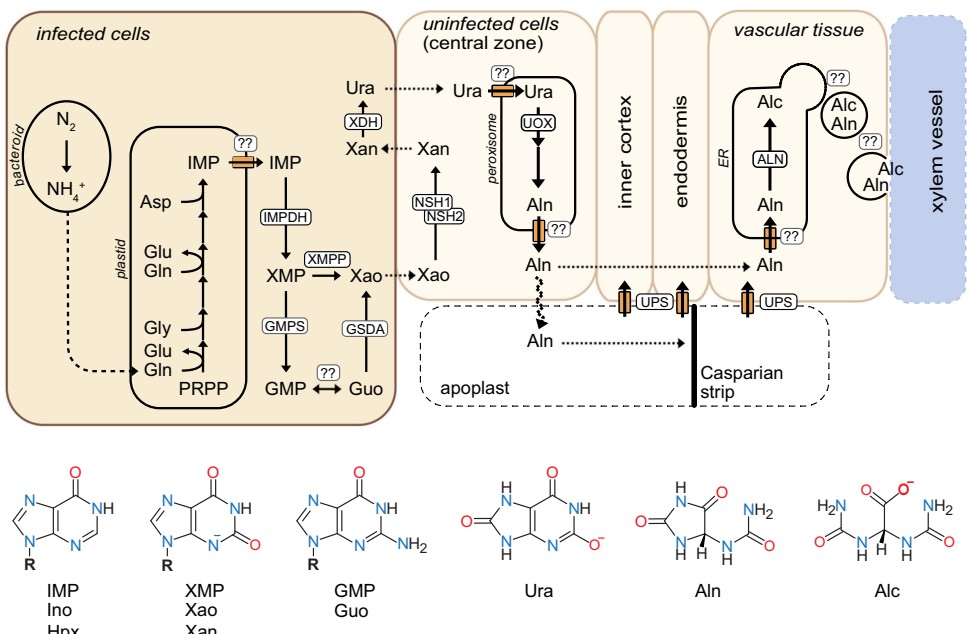

**Fig. 7 | Updated model for ureide biosynthesis and export in bean nodules.** The model integrates our experimental data with the current knowledge but also includes some speculative aspects. In particular the way allantoin (Aln) and allantoate (Alc) are exported from the ER to the xylem vessels in the vascular tissue is unknown. Here, a vesicle transport has been suggested, but transporter-mediated export through the cytosol may occur instead. It is also not clear if Aln and maybe in part also Alc are transported mainly through the symplast or the apoplast from the central infected zone to the vascular tissue. We assume that both pathways play a role, although a proliferation of tubular ER and its close association to peroxisomes in the uninfected cells[51] suggest that the transport mainly occurs symplastically in the ER. Furthermore, it is unknown whether the inner plastid membrane contains an IMP exporter and how uric acid (Ura) enters the peroxisome and Aln leaves the peroxisome and enters the ER. The ER entry is here shown in the vascular tissue but may already occur in the uninfected cells of the central zone, where the tubular ER is strongly proliferated[26, 28, 51]. Whether xanthosine (Xao), xanthine (Xan) and Ura diffuse through plasmodesmata between infected and uninfected cells or whether transport proteins are involved is unknown as well. Figure created with Affinity Designer by Serif. Metabolites: PRPP phosphoribosyl pyrophosphate, IMP inosine monophosphate, AMP adenosine monophosphate, Ino inosine, Hpx hypoxanthine, XMP xanthosine monophosphate, Xao xanthosine, Xan xanthine, GMP guanosine monophosphate, Guo guanosine, Ura urate, Aln allantoin, Alc allantoate. Enzymes: IMPDH IMP dehydrogenase, GMPS GMP synthetase, XMPP XMP phosphatase, GSDA guanosine deaminase, NSH1 and NSH2 nucleoside hydrolases 1 and 2, XDH xanthine dehydrogenase, UOX urate oxidase, ALN allantoinase, ER endoplasmic reticulum. 'R' in the chemical formulas is either a proton for the nucleobases (Hpx, Xan) or ribose for the nucleosides (Ino, Xao, Guo) or ribose-5-phosphate for the nucleotides (IMP, XMP, GMP).

up the coding sequence of the next gene upstream (sequences in the Source Data file). The amplicons were ligated into pY2CS-mNeonGreenSKL (V197) via *Asc*I and *Xho*I replacing the 35 S promoter. This resulted in the clones pY2CpXMPP-NeonGreenSKL (H659), pY2CpGSDA.1-NeonGreenSKL (H660), pY2CpGSDA.2-NeonGreenSKL (H661), pY2CpGSDA.3-NeonGreenSKL (H662), pY2CpNSH1-Neon-GreenSKL (H663); pY2CpNSH2-NeonGreenSKL (H664), pY2CpXDH-NeonGreenSKL (H665), pY2CpUOX-NeonGreenSKL (H666), pY2Cp-ALN-NeonGreenSKL (H667). A longer version of the *XDH* promoter was amplified with the primer pair P3085 and P3088, introducing flanking *Asc*I and *EcoR*I restriction sites. The amplicon was ligated into V197 via *Asc*I and *EcoR*I, resulting in the clone pY2CpXDHlong-mNeonGreenSKL (H1516).

To make more versatile binary vectors for transient expression and transformation pY2empty (V196) was opened with *Hind*III, the overhangs blunted and the vector religated. This step eliminated a *Hind*III site creating pY3empty (V200). A phosphinotricin resistance (PAT) gene cassette was built using pAMPAT-MCS (NCBI accession no. AY436765) as template by amplifying a nopaline synthase promoter-PAT gene fragment flanked by *Asc*I and *Kpn*I sites with primers P1677 and P1678 and a nopaline synthase terminator fragment flanked by *Kpn*I and *Mlu*I with primers P1679 and 1680, respectively, and assembling these fragments to create a (*Asc*I)-pNOS-PAT-(*Kpn*I)-NOSt-(*Mlu*I) construct in pJet1.2 (H742). This *Asc*I-*Mlu*I fragment was cloned into the *Asc*I site of pY3empty (V200) such that transcription is directed towards the left border sequence. The resulting vector pY3B (V201, documented in Source Data File) was digested with *Asc*I and *Pac*I and

an insert fragment containing the mNeonGreenSLK expression cassette from pY2CS-mNeonGreenSKL (V197) generated with the same enzymes cloned in, generating pY3B-CS-mNeonGreenSKL (V202). In this vector the expression cassette was excised with *Asc*I *Xba*I and replaced with the expression cassette from pXNS2pat-Strep[64] (V42) which allows to express a protein as N-terminal Strep-tag fusion, creating pY3B-NS2-Strep (V209). To generate a vector allowing to express a protein as N-terminal myc-tag fusion, pY3B-NS2-Strep (V209) was digested with *Asc*I and *Xma*I and the expression cassette from pXNS2pat-myc[17] (V103) inserted, that had been excised with the same enzymes. This vector was named pY3B-NS2-myc (V242).

From cDNA generated by reverse transcription of nodule RNA from *P. vulgaris* cv. Negro Jamapa, *XMPP* was amplified the primers P2561 and P2562 introducing *Cla*I and *Xma*I restriction sites and cloned into pXCScpmv-HAStrep[60] (V69) allowing for expression of C-terminal HA-Strep-tagged XMPP in planta (clone pXCScpmv-XMPP-HAStrep, H1235). From the same cDNA, *NSH1* and *NSH2* were amplified with primer pairs P2789 and P2596, and P2790 and P2791, respectively, introducing flanking *Nco*I and *Xma*I restriction sites and cloned into pJET1.2 (Thermo Fisher). To eliminate two internal *Nco*I sites in *NSH1*, native *NSH1* sequence was replaced with a synthetic DNA fragment (Supplementary Table 1) using *Nde*I and *Xma*I sites. To eliminate internal *Nco*I and *Cla*I sites, cloned *NSH2* was amplified again with P2792 and P2791 and re-cloned. Note that there was still a different *Nco*I site present in *NSH2* which could not be predicted from the genome sequence due to an allelic difference (Source Data file). Nonetheless, for expression in planta, the cDNAs of *NSH1* and *NSH2* were cloned via *Nco*I and *Xma*I into

pY3B-NS2-Strep (V209) and pY3B-NS2-myc (V242) generating pY3B-NS2-NSH1-Strep (H1339), pY3B-NS2-NSH1-myc (H1340), pY3B-NS2-NSH2-Strep (H1342) and pY3B-NS2-NSH2-myc (H1342).

## Protein purification and determination of kinetic constants, immunoblots

N-terminal Strep- (and myc-) tagged variants of XMPP, NSH1 and NSH2 from *P. vulgaris* were affinity purified after transient expression in *N. benthamiana*. For protein purification, 0.75 g of Agrobacteria infiltrated leaves were ground in 1.5 ml buffer E containing 100 mM HEPES (pH 8.0), 100 mM NaCl, 5 mM EDTA (pH 8.0), 0.005% Triton X-100, 10 mM dithiothreitol, 1:625 diluted Biolock (IBA Life Sciences), 1:10 diluted protease inhibitor (complete protease inhibitor cocktail, Roche). After centrifugation, 40 μl of StrepTactin Macroprep (IBA Life Sciences) was added to the supernatant and incubated for 10 min rotating at 4 °C. The mixture was washed three times with 1 ml buffer W containing 100 mM HEPES (pH 8.0), 100 mM NaCl. 0.5 mM EDTA (pH 8.0), 0.005% Triton X-100, 2 mM dithiothreitol, and centrifuged at 700 g for 30 s between wash cycles.

XMPP activity was determined usinthe EnzCheck Phosphatase Assay Kit (Thermo Fisher) according to the manufacturer's recommendations, but in a total reaction volume of 0.2 ml. The reaction mix was prepared at room temperature in a 0.1 cm quartz cuvette. After the addition of 10 μl purified XMPP (0.4 μg), the mixture was incubated for 10 min to allow the coupling system to remove any contaminant Pi. The reactions were initiated by the addition of substrate and the reaction rates were measured as change of absorption over time at 360 nm. Three independent reactions with XMP concentrations of 2.5, 5.0, 10.0, 20.0, 50.0 and 100 μM were performed and the data fitted to the Michaelis-Menten equation using the Prism V4 software (GraphPad).

NSH activities and NSH protein amounts were assessed using a UV/VIS spectrometer (UV-2700; Shimadzu). The reaction mixture was setup at ambient temperature in a 0.5 cm quartz cuvette and a total volume of 0.3 ml reaction buffer containing 50 mM HEPES (pH 8.0) and 0.125 mM xanthosine. After the addition of 10 μl purified NSH1 (0.3 μg) or NSH2 (0.15 μg) or 20 μl of NSH1/NSH2 complex (0.2 μg), the mixture was incubated for 2 min at 22 °C and the decrease in absorption due to the consumption of xanthosine was monitored over time at 248 nm. The specific activities for determination of guanosine inhibition were measured using 0.125 mM xanthosine in the presence of 0, 0.125 or 0.25 mM guanosine. Note, that the NSH1/NSH2 complex used for these measurements was purified via Strep-tagged NSH2 (and not Strep-tagged NSH1) to obtain only heteromeric NSH1/NSH2 complex and no homomeric NSH1/NSH1 complex. All proteins were quantified on Coomassie-stained SDS gels with BSA as standard with an Odyssey Fc Imager (Li-COR).

For immunodetection, a Strep-Tactin alkaline phosphatase conjugate at a dilution of 1:4000 (2-1503-001, IBA Lifesciences) was used for the detection of Strep-tagged proteins and a monoclonal anti c-myc antibody at a dilution 1:400 (11667149001, Roche) with a secondary anti-mouse IgG alkaline phosphatase conjugate antibody (A3526, Sigma) diluted 1:10,000 was used for the detection of myc-tagged proteins.

## Transcriptome data analysis

RNA-seq data were obtained from the NCBI short Read Archive (SRA, accession PRJNA322355 for *M. truncatula*; PRJDB2819 for *L. japonicus*; PRJNA322335 for *P.vulgaris*; PRJNA79597 and PRJNA208048 for *G. max*). Growth conditions and experimental procedures for the datasets are described[44,65–67]. RNA-seq data were processed using the CLC Genomics Workbench (Qiagen, ver. 7.5.5).

## Cas9 in vitro cleavage assay

The in vitro transcribed guide array (300 ng μl⁻¹) was incubated with 1:20 diluted SpyCas9 (NEB) at ambient temperature. Two controls without either the Cas9 or the guide array were included for subsequent comparison of the DNA template cleavage efficiency. After 10 min, 300 ng of linearized H1493 was added and the reaction mixture was incubated at 37 °C for 20 min. The reaction was stopped by addition of Proteinase K and the samples were separated on an agarose gel.

## Plant growth and transformation

*Nicotiana benthamiana* plants were cultivated under long-day conditions (16 h of light at 22 °C and 85 μmol m⁻² s⁻¹ light intensity/ 8 h of darkness at 20 °C) at 60% humidity. For transient Agrobacterium-mediated (co-)expression of constructs, bacteria with an optical density of 0.5 at 600 nm were infiltrated into young, fully expanded leaves of 4-week-old *N. benthamiana* plants. Plants of *P. vulgaris* cultivar Negro Jamapa and *Glycine max* cultivar Williams 82 were grown in a growth chamber under a 16 h photoperiod and light intensity of 125 μmol photons m⁻² sec⁻¹ at 28 °C during the day and 25 °C during the night (65% humidity). The bean and soybean plants were cultivated in a vermiculite:perlite mixture (2:1) or in Seramis (Westland Deutschland GmbH) in 12 cm diameter pots and were watered three times a week with B&D nutrient solution[68]. Each time, 200 ml of accordingly diluted nutrient solution was used per pot and plant.

For transformation, seeds were surface sterilized in 70% ethanol for three minutes and washed three times (2 min each) with distilled water. Sterilized seeds were placed in a wet vermiculite:perlite mixture (2:1) for germination and grown for four days until they developed a hook at the cotyledonary node. *Agrobacterium rhizogenes* K599 harboring a CRISPR or promoter:reporter-gene constructs were used to inoculate the seedlings. In detail, a needle tip covered with bacteria was used to wound but not pierce the plant three times at the cotyledonary node. Two to three weeks after transformation non-fluorescent roots were removed from the infection site. At that time, plants were inoculated with *Rhizobium tropici* CIAT899 by adding them to the nutrient solution. Mature nodules from transgenic roots identified by GFP or mNeonGreen expression were harvested after 4 weeks and either stored at −80 °C for metabolite analysis or directly processed to generate cross sections.

## Amplified Fragment Length Polymorphism

Genomic DNA was extracted from freeze dried nodules[69]. For amplified fragment length polymorphism (AFLP), the ABI Prism 310 Genetic Analyzer (Thermo Fisher) was used for the detection of CRISPR editing events[61]. Amplicon sizes of the predicted sgRNA target site from wild type and potentially edited plants were compared. Fragments including the sgRNA binding region were amplified from genomic DNA of CRISPR transformed and wild type nodules with the following primer combinations: P2521 and P2498 (for *XMPP*); P2499 and P2500 (*XDH*); P2501 and P2502 (*GSDA.1*); P2503 and P2504 (*GSDA.2*); P2505 and P2506 (*GSDA.3*). Fluorescent primers, labeled with JOE (Merck Millipore) or 6-FAM (Merck Millipore), were used as third primer in the CRISPR and WT reactions, respectively. PCRs were performed using a two-step protocol (95 °C, 3 min; first step: [95 °C, 15 s; 60 °C, 15 s; 72 °C, 1 min; 25 cycles]; second step: [95 °C, 15 s; 52 °C, 15 s; 72 °C, 1 min; eight cycles]; 72 °C, 3 min). For analysis, 1 μl of the sample PCR and 1 μl of the corresponding wild type reaction were mixed with 0.2 μl of Orange 500 DNA Size Standard (NimaGen) in 10 μl Hi-Di formamide (Thermo Fisher) and incubated at 95 °C for 5 min before the run.

## Sample preparation and metabolite analysis

The method was adapted from Hauck[70]. All nodules from a potentially transgenic root, identified by GFP fluorescence, were harvested and freeze dried. Nodule material was homogenized with one 10 mm steel bead using a mixer mill MM 400 (Retsch) for 3 min at 28 s⁻¹ in a precooled 2 ml plastic centrifuge vial. After homogenization, 1 ml of extraction buffer (10 mM NH₄Ac, pH 7.5; 60 °C) was added to 1 mg DW

nodules. Samples were incubated for 10 min at 95 °C while shaking at 1000 rpm in a centrifuge vial shaker and then cooled for 5 min on ice. After centrifugation at 20,000 xg and 4 °C for 10 min, 900 µl of the supernatant was transferred into a fresh reaction tube. Samples were centrifuged again at 55,000 xg and 4 °C for 15 min to further remove contaminants and transferred to a new tube.

XMP, xanthosine, xanthine, guanosine, guanine, inosine, hypoxanthine and allantoate were quantified using an Agilent HPLC 1200 system with a Polaris 5 C18-A 50 × 4.6 mm column (Agilent Technologies) coupled to an Agilent 6460 C series triple quadrupole mass spectrometer. Measurements were performed in positive mode. Ammonium acetate (10 mM, pH 7.5) and 100% methanol served as solvents A and B. The gradient was: 0 min, 5% B; 1.5 min, 5% B; 3.5 min, 15% B; 6 min, 100% B; 7 min, 100% B, 7.1 min, 5% B; and 13 min, 5% B. The flow rate was 0.8 ml min$^{-1}$ and the injection volume was 20 µl. Nodule extracts were diluted 100-fold for the quantification of xanthine in *xdh* nodules. Allantoin was quantified using a 150 × 2.1 mm SeQuant ZIC-cHILIC (3 mm, 100 Å; Merck Millipore). Solvent A was acetonitrile with 50 mM ammonium acetate, pH 5.8, in a ratio of 95:5 and solvent B was water, 50 mM ammonium acetate, pH 5.8; and acetonitrile in a ratio of 50:45:5, respectively. For HILIC chromatography, the nodule extracts were freeze-dried overnight and dissolved in HILIC solvent A using 1.5-fold the original volume. The injection volume was 10 µl and the flow rate was 0.3 ml min$^{-1}$. The gradient was: 0 min, 0% B; 5 min, 5% B; 10 min, 25% B; 15 min, 30% B; 17 min, 65% B; 20 min, 95% B; 20 min, 95% B; 31 min, 95% B; 31.1 min, 5% B; 40.1 min, 5% B. Standard curves for XMP, xanthosine, xanthine, guanosine, guanine, inosine and hypoxanthine were generated with external standards spiked 1:10 into the wild type matrix. For quantification of allantoate and allantoin, external standards were spiked into the *xdh* matrix, because these metabolites were not detectable in this genetic background. MS parameters are listed in Supplementary Table 2. Measurements that did not fit the quality criteria for retention time, qualifier to noise ratio or with a signal to noise ratio below 10 were called 'not detected'. Metabolites were quantified with the MassHunter Quantitative Analysis tool (Ver. B.09.00, Agilent Technologies).

**Confocal laser scanning microscopy**

Longitudinal cross sections of 60 µm from transgenic nodules were prepared with a Leica vibratome VT1000S. During the procedure, nodules were glued onto a plate using super glue and kept in 50 mM NaH$_2$PO$_4$ x Na$_2$HPO$_4$ (pH 7.5). Images of the cross sections were acquired using a Leica TSC SP8 microscope equipped with an HC PL FLUOTAR 10 × 0.30 dry or HC PL APO CS2 × 40 1.10 water immersion objective (Leica, Germany) and processed using the Leica Application Suite X (Leica Microsystems). GFP and mNeonGreen were excited with a laser at 488 nm and the emitted fluorescence was collected from 502 to 537 nm. Xanthine fluorescence was collected from 595 to 622 nm after excitation at 552 nm. To prevent crosstalk between GFP and xanthine, images were acquired by sequential scanning.

**Accession numbers**

The genome sequences used in this work can be found with the following locus identifiers (*P. vulgaris* v2.1 annotation): *XMPP* (Phvul. 007G056000), *GSDA.1* (Phvul.007G185600), *GSDA.2* (Phvul.009G22 0800), *GSDA.3* (Phvul.003G124100), *NSH1* (Phvul.001G188700), *NSH2* (Phvul.003G000600), *XDH* (Phvul.005G148000), *UOX* (Phvul. 007G234300) and *ALN* (Phvul.006G186700). Because the sequences were obtained from the bean cultivar Negro Jamapa, there are a few sequence differences to published genome sequences. Usually, these differences do not affect the amino acid sequences. The cloned sequences are deposited in the Source Data file.

**Reporting summary**

Further information on research design is available in the Nature Research Reporting Summary linked to this article.

## Data availability

The datasets generated and analyzed as part of this study are included in this article (and its supplementary information files). Mass spectrometry raw data can be supplied upon request Source data are provided with this paper.

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

## Acknowledgements

We thank André Specht and Hildegard Thölke for technical assistance and Markus Niehaus for help with the design and construction of CRISPR vectors as well as Jana Streubel for donating the MoClo compatible shuttle vector. We also thank Marta Santalla (CSIC Pontevedra, Spain) and Javier Ollero (University of Sevilla, Spain) for the Negro Jamapa seeds and the *Rhizobium tropici* CIAT 899 strain, respectively. Jannis Rinne we thank for his assistance in harvesting nodules and technical expertise in AFLP analysis. We also are grateful to Leonie Fischer for contributing to the cloning of the pY vector series, Melina Wehrspohn for her help in preparing cross sections and Helge Küster for access to the vibratome. This work was supported by the Deutsche Forschungsgemeinschaft (DFG) grants WI3411/7-1, WI3411/8-1 and INST 187/741-1 FUGG to C.-P.W.

## Author contributions

C.-P.W. devised the project and supervised the work. C.-P.W., N. M-E., L.V. and K.J.H., designed the experiments and interpreted the results. L.V. created most clones and the CRISPR mutants in bean as well as bean nodules expressing promoter-reporter constructs. K.J.H created the XDH CRISPR mutants in soybean. K.J.H and L.V. performed the metabolite analyses and L.V. characterized the XMPP and NSH enzymes biochemically. N.M.E. and L.V. generated the promoter clones, cross sections and carried out the confocal microscopy. M.H. compiled and analyzed gene expression data. L.V performed statistical analyses. L.V. and C.-P.W. wrote the manuscript, which was revised by all authors.

## Funding

## Competing interests

The authors declare no competing interests.
