## [Peer Review File · Nature Communications]

Enzymes and cellular interplay required for flux of fixed nitrogen to ureides in bean nodulesREVIEWER COMMENTS

Reviewer #1 (Remarks to the Author):

The authors have presented a carefully prepared manuscript that provides new information on both the biochemical pathways that are involved in the generation of ureides for transport of fixed nitrogen in tropical legumes as well as distribution of these enzymatic activities in infected and uninfected cells of nodules. The majority of data are presented for common bean, an appropriate model, with supporting information from soybean. This pathway is critically important in the metabolic conversion of fixed nitrogen for long distance transport, but is under-studied in this context in plants because this role is restricted to tropical legumes. The authors have done a good job of integrating the information from studies on legumes and other models in preparing and presenting their work in a way that will be of interest to a wider range of readers.

All the methodology is sound and described in a manner that could be replicated with little or no extra information. The use of a transcriptional meta-analysis at the outset of the manuscript is a conceptually simple and elegant way for the authors to test their hypotheses and focus their efforts on genes likely to play an important role in the nodules. The identification of the Phaseolus XMPP and characterization is clearly and succinctly described. From here, a genetic approach using CRISPR derived mutants in hairy roots and metabolite analysis in nodules is informative and convincing, and the conclusions drawn are supported by the data. Promoter-reporter analysis in nodules then allows the authors to partition the enzymatic activities among the infected and uninfected cells of the nodules, resulting in an updated model of the provision of xanthine for ureide synthesis and export. In some cases, these are hampered by low levels of expression from the endogenous promoters based on the images provided. It is not clear to me whether the authors considered and rejected the idea that genes might be induced in some, but not all, infected or uninfected cells based on proximity to each other. This is addressed more specifically in a comment below. In my opinion the authors should be commended on the clear presentation of their work. Specific comments, questions, and suggestions are listed below.

The introduction is well written and focused on relevant literature. I will suggest minor editorial changes, including limiting the use of "especially" and "in any case" (page 2, paragraph 2). I also suggest rewording "probably occurs mainly" (page 3, paragraph 2). To me, it is difficult to determine if the authors have confidence in the published results with this phrase.

In the results section, I suggest changing "XMPP is required but not indispensable" to "involved but not indispensable" (p. 7) as it is clearer. It is not clear how the authors are defining "required."

Can the authors comment on the general appearance of the *xdh* mutant nodules in Phaseolus. In soybean they appear much smaller, with a lack of leghemoglobin with apparently fewer infected cells (Supp Fig. 4)

Page 9 – can the authors comment why they are not considering the lack of detectable hypoxanthine as a difference in claiming that *nsh1* metabolite contents did not differ from wild type control? It appears that the control may have had one or more samples that were not detectable, leaving the value possibly not significantly different than zero, but it appears that hypoxanthine was detected in one or two samples. This acknowledgement in no way would weaken the conclusions.

Based on Figure 5 and Supplementary Figure 7, it appears that there may be XDH expression in uninfected cells immediately adjacent to or interspersed with the infected cells. In Figure 6, this is depicted as being in only infected cells. Is there enough data from additional images to rule out *XDH* expression in both infected and uninfected cells?

The authors make an interesting suggestion as to why NSH1/2 are not in infected cells because of

guanosine concentrations, however the suggestion of urate's role as a local ROS/RNS scavenger would not need to be exclusive to infected cells. Such a mechanism operating in both infected and uninfected cells at the boundary between them could be an equally effective mechanism.

Early in the paper the authors used a different color or question marks to clearly indicate to the reader where there are insufficient data to draw a conclusion on an enzyme's location and activity. In Figure 6, the movement of allantoin and allantoate through a membrane bound vessel to the xylem is not indicated in this way, though the authors state it is purely speculative. I do not disagree with the possibility/likelihood of the proposed pathway, but I suggest changing Figure 6 so that the speculative nature of this is clearly apparent, perhaps also showing allantoin and/or allantoate moving through the cytosol through as yet unidentified transporters, similar to yet unidentified enzymes or transporters depicted in Supplemental Figure 1.

Thank you for the opportunity to review this work.

Christopher Todd
University of Saskatchewan

Reviewer #2 (Remarks to the Author):

Manuscript#: NCOMMS-22-12598, entitled: "Enzymes and cellular interplay required for the flux of fixed nitrogen to ureides in bean nodules, is a work interesting and sound for plant biologists. It shows new information that updates a fundamental route in tropical climate legumes such as beans and soybeans, as the synthesis of purine nucleotides and ureides in the nodules of these plants.

Methods and procedures are well explained in the manuscript (see below), and it shows the usefulness of CRISPR/Cas9 mutations in transformed hairy roots of common bean. This is a powerful analytical tool to unveil the specific role of genes or proteins within an enzymatic pathway, such as ureide synthesis in the nodules of these legumes. Moreover, the generation of promoter-marker fusions and expression of these constructs in nodules is used to demonstrate the induction of expression and to locate the encoded proteins of these genes in either the infected or uninfected cells of the nitrogen-fixing nodules.

In general, the results are clear and well-presented and the conclusions seem overall well supported. There are, however, a couple of flaws that merit further attention.

First concerns the correlation between expression level and functional implication of each gene in ureide synthesis in nodules, by using publicly available data. The rationale for this correlation is fine, since expression in nodules versus the one in roots is compared among ureidic and non-ureidic legumes, and thus, it seems clear that high expression in nodules of ureidic legumes should be expected for those genes that will have a role in ureide production. However, the data in Table 1 are, somehow, simplified in a way that could result in a misunderstanding by the scientific community.

Many of the enzymes in the purine synthesis pathway have been only characterized at the biochemical level, and there are several enzymes in the list that are coded by more than one gene in the common bean genome, and for many of them, the actual participation of each gene copy is up to now unknown. This is for example the case of PRS, Pur1PRAT, Pur4 FGARAT and ALN. There are three copies coding PRS enzyme in the bean genome (and the number is higher in soybean), but the actual participation of none of them in legume nodules has been established. The assignment of one gene to IMP synthesis in nodules of common bean or soybean is, at least, arbitrary (even if the expression level of the gene copies is the highest one in nodules). In the case of Pur1 (PRAT), there are also three gene copies, although in this case the functional specialization of one particular gene (the one mentioned in the table) has been established in common bean (Coletto et al. 2016). The third case is even more problematic since there are two genes in the common bean genome and both of them are expressed in nodules to almost similar levels (according to publically available expression data). In the

case of ALN it is also not clear, since the two genes coding ALN from common bean are expressed in nodules. This also happens for two of the four ALN copies in soybean, although in soybean it looks like the two genes expressed in nodules are not expressed elsewhere. Therefore, I believe that to avoid confusion, the data in Table 1 must be completed to make it clear which information has been verified and which one is speculative, or just inferred from expression levels data (in this study). Therefore a heading or subheading should be included in the functional assignment to inform about this.

The second, and perhaps more relevant, concerns the localization of XDH in infected cells. According to the expression results of a fusion of the XDH promoter to a fluorescent marker XDH is unexpectedly located in the infected cells (bear in mind that xanthine, the substrate of the enzyme, accumulates in non-infected cells, according to this and previous work).

According to the information provided as supplementary material, the region of promoter sequence used in this case is too short (748 bp) for any plant promoter analysis. In the methods, the authors state that 3Kb regions of the corresponding promoters or length up to the next coding region are used, but this seems not to be the case for XDH. Moreover, since the proposed XDH location in infected cells is surprising, to say the least, I believe that it can not be assigned just by the results of this short promoter region. Authors should be aware that there are very few cis-regulatory motives within this small region, while there are a couple of possible regulatory sequences (including several related to anaerobiosis responses) upstream of this region. The use of promoter activation could give wrong results if the motives are not present within the promoter fragment used, instead, basal expression (that for XDH is expected to occur in all cell types) could be observed. Authors should try a longer promoter region or another way of protein location analysis to further confirm their new and quite surprising results

The last concern is about the method for both the construct and the characterization of CRISPR mutants. Hairy roots transformation and CRISPR/Cas9 edition or mutation are quite novel techniques, not shown previously in common bean. Thus the information on these methods should be as accurate as possible. Concerning the constructs to mutate each of the target genes, the authors use golden gate technology to assemble all components of the expression vector, and this is explained in all required detail. However, there is very little, or no information at all, about the design of the guides, where are the target sequences within the genomic regions of each gene?; are there any predicted off-targets sequences?; to which exon are they directed?, which program was used to find the PAM sequences? and so on. Methods explain only that they were prepared according to Xie et al, 2015. However, there is still some important information that should be provided. For example, the cited method is recommended to boost the targeting by the cloning of several guides into a polycistronic construct. However, according to the authors, they used single RNA guides, for each gene (even in the case of targeting three genes, it seems that there is only one guide per gene). I recommend supplying this information within the supplemental data. There is also a surprising lack of analysis of the expression level of the mutants. Enzymatic activity or accumulation of metabolites associated with the lack of activity is, for sure, a measure of the mutation effect, but gene expression using specific primers could help to discriminate among each mutant effect, at least in those cases in which more than one gene could be implicated, as in the NSH or in the GSDA activities, but also in the effects of mutation in nodules of XDH.

In summary, and although I believe that the results presented in the manuscript are sound enough to be published, there are several aspects that should be improved before I would recommend publication in this journal. I do recommend acceptance after major changes has been done.

Please find below the responses to the reviewers' comments.

Reviewer #1:

COMMENT: The introduction is well written and focused on relevant literature. I will suggest minor editorial changes, including limiting the use of "especially" and "in any case" (page 2, paragraph 2). I also suggest rewording "probably occurs mainly" (page 3, paragraph 2). To me, it is difficult to determine if the authors have confidence in the published results with this phrase.

REPLY: We agree with the proposed changes on page 2 and have incorporated them into the text. On Page 3 we would like to leave it as it is ("Purine de novo synthesis probably occurs mainly in infected cells of cowpea nodules"), because the confidence one would like to have in this case is not really supported by the literature. Although it makes sense to assume that purine biosynthesis is much stronger in infected cells than in uninfected cells, so far the experimental evidence is not sufficient to be very confident about this, in our opinion.

COMMENT: In the results section, I suggest changing "XMPP is required but not indispensable" to "involved but not indispensable" (p. 7) as it is clearer. It is not clear how the authors are defining "required."

REPLY: We agree with the proposed changes and have incorporated them into the text.

COMMENT: Can the authors comment on the general appearance of the *xdh* mutant nodules in *Phaseolus*. In soybean they appear much smaller, with a lack of leghemoglobin with apparently fewer infected cells (Supp Fig. 4).

REPLY: Similar to soybean, *Phaseolus XDH* mutant nodules also appeared smaller and less red/ pink, likely due to the lack of leghemoglobin. However, we did not document this phenotype to the same extent as in soybean. We added a sentence to the legend of Supplementary Figure 4a: "*P. vulgaris* nodules lacking *XDH* showed a similar phenotype (not shown)."

COMMENT: Page 9 – can the authors comment why they are not considering the lack of detectable hypoxanthine as a difference in claiming that *nsh1* metabolite contents did not differ from wild type control? It appears that the control may have had one or more samples that were not detectable, leaving the value possibly not significantly different than zero, but it appears that hypoxanthine was detected in one or two samples. This acknowledgement in no way would weaken the conclusions.

REPLY: We are claiming that *nsh2* (not *nsh1*) metabolite contents did not differ from the wild type. There was generally very little hypoxanthine (< 10 nmol / g DW) in the samples irrespective of the genotype, very close to the detection limit. Because we did not detect hypoxanthine in *nsh2* at all, we cannot say with any certainty whether *nsh2* contained even less hypoxanthine. Even if that were the case, the total amounts are so small that they are insignificant compared to, for example, xanthine, which was at least 20-fold more abundant.

COMMENT: Based on Figure 5 and Supplementary Figure 7, it appears that there may be *XDH* expression in uninfected cells immediately adjacent to or interspersed with the infected cells. In Figure 6, this is

depicted as being in only infected cells. Is there enough data from additional images to rule out XDH expression in both infected and uninfected cells?

REPLY: We repeated the experiment with the originally chosen *XDH* promoter (a 735 bp region between the stop codon of the upstream gene and the *XDH* start codon) and also with a reporter construct containing a longer 3 kb version of the *XDH* promoter (as requested by reviewer #2), which also includes part of the upstream gene. The data is shown in the new Figure 6. Irrespective of promoter fragment length, the reporter was almost exclusively located in infected cells. Rarely, there were fluorescent signals which apparently stemmed from neighboring uninfected cells. We attribute these signals to the nature of the samples, which were 60- μ m nodule slices without fixation of the tissue. A small amount of fluorescence may have been displaced during cutting, or in some cases fluorescence from underlying cell layers may have been detected. Nevertheless, the experiments make it clear that *XDH* promoter activity is strongly associated with infected cells, suggesting that XDH is mainly, if not exclusively, expressed there.

In the result section we have added a few lines to report on the new experiments (repeat of XDH promoter study with 735 bp promoter and new study with 3 kb promoter) and have added a sentence offering an explanation for the occasional signals in uninfected cells.

COMMENT: The authors make an interesting suggestion as to why NSH1/2 are not in infected cells because of guanosine concentrations, however the suggestion of urate's role as a local ROS/RNS scavenger would not need to be exclusive to infected cells. Such a mechanism operating in both infected and uninfected cells at the boundary between them could be an equally effective mechanism.

REPLY: In our discussion we try to make exactly this point. Because xanthine diffuses from uninfected to infected cells, urate is made in infected cells - but at the cell periphery - and then diffuses back to the uninfected cells, thereby protecting the boundary zone between both cells from ROS/RNS. Our discussion of this point ends with "Thus, urate production by XDH at the interface between infected and uninfected cells may create a sink for ROS/RNS in this zone of high ROS production and low ROS tolerance". We talk about a "zone", and the interface between infected and uninfected cells. In this model both cell types profit from urate as ROS/RNS sink, but since infected cells have to be especially protected from oxygen but also require oxygen-dependent metabolism, the benefit of urate is probably greater there.

COMMENT: Early in the paper the authors used a different color or question marks to clearly indicate to the reader where there are insufficient data to draw a conclusion on an enzyme's location and activity. In Figure 6, the movement of allantoin and allantoate through a membrane bound vessel to the xylem is not indicated in this way, though the authors state it is purely speculative. I do not disagree with the possibility/likelihood of the proposed pathway, but I suggest changing Figure 6 so that the speculative nature of this is clearly apparent, perhaps also showing allantoin and/or allantoate moving through the cytosol through as yet unidentified transporters, similar to yet unidentified enzymes or transporters depicted in Supplemental Figure 1.

REPLY: We agree with the suggested changes and have highlighted the speculative elements in Figure 6 with question marks. We have also improved the figure legend, mentioning that alternatively allantoin/allantoate transport may also occur through the cytosol via transporters.

Reviewer #2:

COMMENT: First concerns the correlation between expression level and functional implication of each gene in ureide synthesis in nodules, by using publicly available data. The rationale for this correlation is fine, since expression in nodules versus the one in roots is compared among ureidic and non-ureidic legumes, and thus, it seems clear that high expression in nodules of ureidic legumes should be expected for those genes that will have a role in ureide production. However, the data in Table 1 are, somehow, simplified in a way that could result in a misunderstanding by the scientific community. Many of the enzymes in the purine synthesis pathway have been only characterized at the biochemical level, and there are several enzymes in the list that are coded by more than one gene in the common bean genome, and for many of them, the actual participation of each gene copy is up to now unknown. This is for example the case of PRS, Pur1PRAT, Pur4 FGARAT and ALN. There are three copies coding PRS enzyme in the bean genome and the number is higher in soybean), but the actual participation of none of them in legume nodules has been established. The assignation of one gene to IMP synthesis in nodules of common bean or soybean is, at least, arbitrary (even if the expression level of the gene copies is the highest one in nodules). In the case of Pur1 (PRAT), there are also three gene copies, although in this case the functional specialization of one particular gene (the one mentioned in the table) has been established in common bean (Coletto et al. 2016). The third case is even more problematic since there are two genes in the common bean genome and both of them are expressed in nodules to almost similar levels (according to publically available expression data). In the case of ALN it is also not clear, since the two genes coding ALN from common bean are expressed in nodules. This also happens for two of the four ALN copies in soybean, although in soybean it looks like the two genes expressed in nodules are not expressed elsewhere. Therefore, I believe that to avoid confusion, the data in Table 1 must be completed to make it clear which information has been verified and which one is speculative, or just inferred from expression levels data (in this study). Therefore a heading or subheading should be included in the functional assignation to inform about this.

REPLY: We agree that Table 1 is somewhat simplified and that this may lead to confusion. Therefore, we have included a new footnote: “We used the induction of at least one paralog as an indicator of involvement in ureide biosynthesis. Therefore, in several cases not all paralogs are listed, but only those induced in nodules versus roots. Non-induced paralogs may also contribute to ureide biosynthesis.”

For the purpose of this study, we think it is sufficient to list only the induced paralogs, because if at least one paralog is induced, it indicates its involvement in ureide biosynthesis*. Our intention was to identify such genes and not to list all paralogs. Nevertheless, in some cases we have listed all the paralogs: for genes that could be assumed to be involved in ureide biosynthesis, but none of the paralogs show induction. This lack of induction of all paralogs is taken as an indication, that the respective genes are not involved in ureide biosynthesis.

*our experimental data regarding the GSDA paralogs shows that GSDA.2 is induced but is not the only paralog involved in guanosine degradation in nodules. The non-induced paralog GSDA.1, which we could not mutate, is apparently also involved. This shows that non-induced paralogs may be important. Nonetheless, can the induction of genes be used as a tool to identify paralogous groups involved in ureide biosynthesis.

COMMENT: The second, and perhaps more relevant, concerns the localization of XDH in infected cells. According to the expression results of a fusion of the XDH promoter to a fluorescent marker XDH is unexpectedly located in the infected cells (bear in mind that xanthine, the substrate of the enzyme, accumulates in non-infected cells, according to this and previous work). According to the information provided as supplementary material, the region of promoter sequence used in this case is too short (748 bp) for any plant promoter analysis. In the methods, the authors state that 3Kb regions of the corresponding promoters or length up to the next coding region are used, but this seems not to be the case for XDH. Moreover, since the proposed XDH location in infected cells is surprising, to say the least, I believe that it can not be assigned just by the results of this short promoter region. Authors should be aware that there are very few cis-regulatory motives within this small region, while there are a couple of possible regulatory sequences (including several related to anaerobiosis responses) upstream of this region. The use of promoter activation could give wrong results if the motives are not present within the promoter fragment used, instead, basal expression (that for XDH is expected to occur in all cell types) could be observed. Authors should try a longer promoter region or another way of protein location analysis to further confirm their new and quite surprising results.

REPLY: We agree that the selected sequence is relatively short for a promoter, but this length was chosen because in all bean genomes deposited in the Phytozome V13 database, a gene encoding a phosphate acyltransferase (*PTHR*; locus Phvul.L009443) is annotated at this distance upstream of the *XDH* gene. Therefore, in this case, we chose a shorter promoter region between the *PTHR* stop codon and the *XDH* start codon. Nonetheless, we recognize that the location of *XDH* promoter activity is an important aspect of our manuscript and therefore repeated the analysis including a construct with a 3 kb region upstream of the *XDH* start codon although this includes a substantial 3'-part of the *PTHR* gene. The data is shown in the new Figure 6. See comments to reviewer 1 above.

COMMENT: The last concern is about the method for both the construct and the characterization of CRISPR mutants. Hairy roots transformation and CRISPR/Cas9 edition or mutation are quite novel techniques, not shown previously in common bean. Thus the information on these methods should be as accurate as possible. Concerning the constructs to mutate each of the target genes, the authors use golden gate technology to assemble all components of the expression vector, and this is explained in all required detail. However, there is very little, or no information at all, about the design of the guides, where are the target sequences within the genomic regions of each gene?; are there any predicted off-targets sequences?; to which exon are they directed?, which program was used to find the PAM sequences? and so on. Methods explain only that they were prepared according to Xie et al, 2015. However, there is still some important information that should be provided. For example, the cited method is recommended to boost the targeting by the cloning of several guides into a polycistronic construct. However, according to the authors, they used single RNA guides, for each gene (even in the case of targeting three genes, it seems that there is only one guide per gene). I recommend supplying this information within the supplemental data.

REPLY: We used the CRISPOR web tool for the sgRNA design and evaluation of potential off-target effects. The sgRNAs were designed to bind in an exon, preferably within the first half of the corresponding transcript to raise the probability of compromising the function. We have now included this information in the method section of the manuscript. The corresponding sgRNA sequence and target sites can be deduced from the Supplementary Table 1. However, we recognize that this may be tedious and provided an additional Supplementary Fig. 9 to visualize the sgRNA binding sites in each gene. The exact sgRNA binding site sequences and the location in the gene are deposited in the Source Data file.

COMMENT: There is also a surprising lack of analysis of the expression level of the mutants. Enzymatic activity or accumulation of metabolites associated with the lack of activity is, for sure, a measure of the mutation effect, but gene expression using specific primers could help to discriminate among each mutant effect, at least in those cases in which more than one gene could be implicated, as in the NSH or in the GSDA activities, but also in the effects of mutation in nodules of XDH.

REPLY: We disagree on this point. We have shown by AFLP analyses that the respective targeted genes were mutated because of a shift in the open reading frame. The corresponding transcript can therefore not be translated into a functional protein. In many cases the (non-functional) transcripts of these genes will probably still be present. Thus, transcript amounts in mutants will often not even be different from the wild type and no information to discriminate the mutants from the wild type would be gained from gene expression analyses. Even if there would be differences, it is clear that transcripts in frame-shift mutants are non-functional, irrespective of transcript abundance.

REVIEWERS' COMMENTS

Reviewer #2 (Remarks to the Author):

Manuscript#: NCOMMS-22-12598A: Enzymes and cellular interplay required for flux of fixed nitrogen to ureides in bean nodules, suggest that ureide biosynthesis in legumes is a metabolic pathway in which, as already know, both infected and uninfected cells of the nodules are implicated, but the authors show that this contribution is far more complex than expected. Authors use protein overexpression, promoter activity, metabolite analysis and targeted mutants by gene edition for the in vivo analysis of the implication of the some of the enzymes of the pathway. In the revised MS, the authors have addressed most of this referee concerns, and the most relevant questions have been appropriately answered and added to the MS. Therefore, I consider that the MS could readily be published in the present form. I give also my congratulations to the authors for their excellent work.